# Ultrasound Improved the Non-Covalent Interaction of β-Lactoglobulin with Luteolin: Regulating Human Intestinal Microbiota and Conformational Epitopes Reduced Allergy Risks

**DOI:** 10.3390/foods11070988

**Published:** 2022-03-29

**Authors:** Titi Wang, Wenmei Chen, Yanhong Shao, Jun Liu, Zongcai Tu

**Affiliations:** 1College of Life Science, National R & D Center for Freshwater Fish Processing, Jiangxi Normal University, Nanchang 330022, China; wttjxnu@163.com (T.W.); chen_9721@126.com (W.C.); 18354286339@163.com (Y.S.); lj_anyi@126.com (J.L.); 2State Key Laboratory of Food Science and Technology, Nanchang University, Nanchang 330047, China

**Keywords:** ultrasound, β-LG, LUT, allergenicity, human intestinal microbiota

## Abstract

The present study aims to investigate the effects of ultrasound on the non-covalent interaction of β-lactoglobulin (β-LG) and luteolin (LUT) and to investigate the relationship between allergenicity and human intestinal microbiota. After treatment, the conformational structures of β-LG were changed, which reflected by the decrease in α-helix content, intrinsic fluorescence intensity and surface hydrophobicity, whereas the β-sheet content increased. Molecular docking studies revealed the non-covalent interaction of β-LG and LUT by hydrogen bond, van der Walls bond and hydrophobic bond. β-LG-LUT complex treated by ultrasound has a lower IgG/IgE binding ability and inhibits the allergic reaction of KU812 cells, depending on the changes in the conformational epitopes of β-LG. Meanwhile, the β-LG-LUT complex affected the composition of human intestinal microbiota, such as the relative abundance of *Bifidobacterium* and *Prevotella*. Therefore, ultrasound improved the non-covalent interaction of β-LG with LUT, and the reduction in allergenicity of β-LG depends on conformational epitopes and human intestinal microbiota changes.

## 1. Introduction

It is well known that cow’s milk is a significant nutritional supplement in life, while also being one of the eight food allergens [1]. Cow’s milk allergies are especially common in infancy and childhood, with symptoms such as vomiting and diarrhea., and can be life-threatening when severe [2]. Allergy protein components in cow’s milk, such as β-lactoglobulin (β-LG), α-lactalbumin and casein, are the main causes of allergies [3]. It was reported that about 82% of people are allergic to β-LG, which greatly limits the suitable population and applied range of milk [4].

β-LG is a 162-amino-acid globular protein with a molecular weight of 18.36 kDa [5]. The antigenicity of β-LG is related to its linear and conformational epitopes. The former is generally composed of five-to-seven consecutive amino acid linear sequences, and the latter is a discontinuous sequence that mainly depends on the tertiary and quaternary structures of β-LG [6]. Based on the principle that the changes in the structure and spatial conformation of allergens lead to the reduced sensitization of allergic epitopes, various processing methods, such as heating, enzymatic hydrolysis, and glycosylation, have been applied to desensitize β-LG [7]. However, these single desensitization techniques may affect the quality and flavor of proteins during the application process and fail to achieve a good desensitization effect [8]. Currently, reducing the antigenicity of proteins in a reversible binding manner through non-covalent interactions has been popular and less studied. Therefore, it is necessary to reduce the antigenicity of the protein by modifying β-LG by sonication combined with non-covalent interactions.

Ultrasound, as a safe and efficient non-thermal processing method, can significantly improve food quality and has been applied in many fields, such as protein modification [9]. Luteolin (LUT), an important flavonoid compound, has been reported to have a variety of benefits on good human health, such as anti-oxidative and anti-tumor [10]. In a recent study, Pu et al. determined that flavonoids can reduce the antigenicity of β-LG through non-covalent interactions [11]. However, there is very little information on the effects of ultrasound combined with non-covalent interaction modification on the allergenicity of β-LG. Based on the technical advantages of ultrasound in improving the quality of proteins [12] and the fact that flavonoid-modified β-LG can reduce its allergenicity, we aim to study the effect of ultrasound on the allergenicity of β-LG-LUT complexes.

Food allergy is also associated with gastrointestinal microbiota [13]. There were significant differences in fecal microbiota between infants with milk allergies and normal infants. For example, Clostridium difficile and Onomatopoeia were more common in the fecal microbiota of infants with milk allergy [14]. Canani et al. showed that formula supplemented with lactobacillus rhamnose can promote tolerance acquisition in infants with milk protein allergy [15]. Ling et al. reported that the changes of fecal microbiota in infants with early milk allergies was related to the pathogenesis of food allergy, and may lead to decreased diversity of microbiota in infancy [16]. Therefore, it is of great significance to explore the relationship between the allergenicity of β-LG-LUT complex and human intestinal microbiota.

In this paper, we prepare a β-LG–LUT complex under ultrasound to investigate the changes of structure, the potential allergenicity and human intestinal microbiota. The structural changes of the β-LG–LUT complex were investigated by multi-spectroscopy and molecular docking. The IgG/IgE binding activity of rabbit polyclonal antibody and serum of milk allergy patients was analyzed by ELISA. The effect on the release of cell biological active mediators, such as histamine and interleukin-6 (IL-6), was analyzed through the KU812 cell model. Finally, the effects of β-LG-LUT complex on human intestinal microbiota represented by feces were studied in vitro via fermentation experiments.

## 2. Materials and Methods

### 2.1. Materials

β-LG were purchased from Sigma Company, (St. Louis, MO, USA); LUT (HPLC ≥ 98%), pepsin and trypsin were obtained from Beijing Solarbio Technology Co., Ltd. (Beijing, China). 8-anilino-1-naphthalene sulfonate (ANS) were obtained from Shanghai Aladdin Biochemical Technology Co., LTD. (Shanghai, China). The VCY-1500 (probe 6 mm) ultrasonic crusher was acquired from Shanghai Yanyong ultrasonic equipment Co., Ltd. (Shanghai, China).

### 2.2. Sample Preparation

β-LG was dissolved in a phosphate buffer (pH = 7.4, 10 mmol/L) to prepare a β-LG solution at a concentration of 5.0 mg/mL and stored at 4 °C. LUT was dispersed in ethanol at a concentration of 2.0 mmol/L and stored at 4 °C for later use. Intermittent ultrasonic treatment was used by a 9 s on and 1 s off pulsation at an actual ultrasonic intensity of 300 W and the time was 15 min [17]. The entire ultrasonic process was cooled by an ice bath, and the sample temperature was kept below 15 °C. Ultrafiltration centrifugation and dialysis were used to remove excess LUT from all samples. The β-LG solution after ultrasonic treatment was named U-LG. The β-LG–LUT complex was simply mixed and named S-LG–LUT. The β-LG–LUT complex after ultrasonic treatment was named U-LG–LUT. β-LG solution without any treatment was named N-LG.

### 2.3. Far-UV Circular Dichroism Spectra

Circular dichroism spectra of β-LG in the range of 190–250 nm were measured according to a reported method [18]. The concentration of β-LG sample was 0.1 mg/mL, the scanning speed was 100 nm/min, and the bandwidth was adjusted to 1 nm. Secondary structure content was analyzed using online analysis software (DichroWeb, http://dichroweb.cryst.bbk.ac.uk/html/process.shtml, access on 16 March 2022).

### 2.4. Fourier Transform Infrared Measurement

The structural changes of β-LG were analyzed by FTIR spectra of DTGS detector [18]. The 2.0 mg sample was determined by FTIR. The spectra were recorded in the range of 1800–1400 cm^−1^ and 32 scans were performed with a resolution of 4 cm^−1^.

### 2.5. Intrinsic Fluorescence Intensity and UV Spectrometry Measurement

The F-7000 fluorescence spectrometer was used to measure the fluorescence intensity of N-LG, U-LG, S-LG–LUT and U-LG–LUT [19]. The measured parameters were: excitation wavelength at 280 nm, emission wavelength range 300–450 nm, voltage 400 V, excitation and emission slit width 5 nm. Hitachi U-2910 UV-visible spectrophotometer was used to analyze the UV absorption spectrum of N-LG, U-LG, S-LG–LUT and U-LG–LUT. The sample was diluted to 1 mg/mL with PBS (pH = 7.4, 10 mmol/L), and the scanning range was 240–400 nm.

### 2.6. Surface Hydrophobicity Determination

The surface hydrophobicity of β-LG was measured by ANS probe method according to Ma et al. [20]. The samples were diluted with PBS (pH = 7.4, 10 mmol/L) to different concentrations (0.065, 0.125, 0.25, 0.5, 1; Unit: mg/mL). The excitation wavelength was 370 nm, the emission wavelength was 400–600 nm, the scanning speed was 240 nm/min, the slit width of excitation and emission was 10 nm, and the voltage was 400 V. The fluorescence intensity was determined by mixing 8 mmol/L ANS solution (pH = 7.4, 10 mmol/L) 20 μL with 4 mL protein solution. Linear regression analysis was used to fit the curve, with protein mass concentration as abscissa and fluorescence intensity as ordinate. The slope of the curve was surface hydrophobicity (H_0_).

### 2.7. Molecular Docking

The interaction between LUT and β-LG was investigational by computational docking simulation using Autodock tools-1.5.6. β-LG molecular model (PDB-ID: 3NPO) was downloaded from the protein database PDB (Protien Date Bank). The three-dimensional structure of LUT molecule was drawn by chembio draw 14 software (CambridgeSoft, Cambridge, MA, USA), then processed by ChemBio 3D Ultra 14.0.0.117 (CambridgeSoft, Cambridge, MA, USA). Lamarckian genetic algorithm was used to calculate the conformation of the calculate ligand and receptor. A total of 100 docking runs were performed, and the conformation with minimum binding energy was selected as the optimal binding site.

### 2.8. Determination of IgG/IgE Binding Ability

The allergenicity of N-LG, U-LG, S-LG–LUT and U-LG–LUT was evaluated by indirect competitive ELISA according to the method of Liu et al. [21] with minor modifications. The binding capacity of IgE was determined by indirect ELISA [22]. The sample was diluted to 10 μg/mL and coated with an enzyme plate. After sealing, the primary antibody (1:50) and secondary antibody (1:200) were incubated. The reaction was terminated 10 min later, and the OD value at 450 nm was read on the microplate reader.

### 2.9. Degranulation of Human Basophil (KU812) Cells

KU812 cells were cultured with 5 × 10^5^ cells/well in 48-well plates. The cells were stimulated with human serum IgE of milk allergy for 24 h and stimulated with N-LG, U-LG, S-LG–LUT and U-LG–LUT at 50 μL/well (1 mg/mL) for 4 h. The secreted components of cells were collected with a sterile centrifuge tube and centrifuged at 1000 RPM for 10 min (cell lysate). The supernatant was collected and the IL-6 and human histamine contents in the supernatant were determined by ELISA kit.

### 2.10. Determination of Human Intestinal Microbiota

Fresh stool samples were provided by three volunteers who had no metabolic or gastrointestinal disease and had not used probiotics, or antibiotics for nearly three months. Stool samples were collected in sterile homogeneous bags and diluted at a ratio of 1:5 (m/v) with 0.1 mol/L sterile phosphate buffer (pH = 7.4). The contents were homogenized and filtered and sorted. Finally, N-LG, U-LG, S-LG–LUT and U-LG–LUT, which simulated gastrointestinal digestion in vitro, were added to homogenized feces for anaerobic culture for 48 h. All operations were carried out under anaerobic sterilization conditions.

### 2.11. High-Throughput Sequencing and Microbial Analysis

After the out-of-population fermentation of human intestinal bacteria, feces were collected by centrifugation and sent to Shanghai Meiji Biomedical Technology Co., Ltd. (Shanghai, China). The primers for sequencing were universal primers for the V4 region of 16S rRNA gene:338F 5′-GTGCCAGCMGCCGCGGTAA-3′ and 806R 5′-GCACTACHVGGGT-WTCTAAT-3′. Then, the samples were collected on the platform provided by I-Sange, a branch of Shanghai Meiji Biomedical Technology Co., Ltd. (Http://www.isanger.com/index.html, accessed on 16 March 2022), for all biological information analysis.

### 2.12. Statistical Analysis

All experiments were repeated three times, and all results are expressed as mean ± standard deviation. SPSS 22.0 software was used for one-way ANOVA and Origin 9.6 software was used for plotting.

## 3. Results and Discussion

### 3.1. Infrared Spectral Analysis

FTIR can be used to determine the molecular structure and chemical bond changes of β-LG. As shown in Figure 1A, compared with N-LG, no new FTIR spectral signal appeared after adding LUT, indicating that LUT was non-covalently bound to β-LG [11]. The most important FTIR absorption bands for proteins are the amide I and amide II bands. The characteristic absorption band of amide I appears at 1600–1700 cm^−1^ (mainly C=O stretching), and the absorption band of amide II appears at 1500–1600 cm^−1^ (C-N stretching combined with N–H deformation vibration) [23]. Compared with N-LG, the peak position of U-LG in the amide I band is shifted from 1652.692 cm^−1^ to 1648.839 cm^−1^. The results indicated that ultrasound alters the C=O stretch of β-LG [11], possibly affected the interaction of β-LG–LUT. S-LG–LUT blue-shifted from 1652.692 cm^−1^ to 1646.911 cm^−1^ in the amide I band, and blue-shifted from 1540.845 cm^−1^ to 1538.916 cm^−1^ in the amide II band. The results suggested that electrostatic interactions occur between β-LG and LUT, and the LUT modification altered the C–N stretch of β-LG [24]. Compared with S-LG–LUT, U-LG–LUT moved from 1646.911 cm^−1^ to 1643.054 cm^−1^. These results suggest that ultrasound affected the structural changes of the β-LG–LUT complex, which may affect its antigenicity.

### 3.2. Circular Dichroism Analysis

The circular dichroism of proteins can reflect their structural changes. As can be seen from Table 1, S-LG–LUT exhibited a lower α-helix content and higher β-sheet and β-turns contents compared with those of N-LG. α-helix of S-LG–LUT decreased to 2.39%, and β-sheet increased to 42.37%, indicating that part of the α-helix of β-LG may be extended to β-sheet whereas β-LG binds with LUT. The change was consistent with other research results [25,26]. However, the secondary structure change of U-LG–LUT is the most significant in all tested samples, for instance, the α-helix content decreased from 13.24% to 0.61% and the β-sheet content increased from 29.14% to 43.66%. Studies have shown that the inhibitory effects of flavonoids on the allergenicity of β-LG were closely related to the decline in the α-helix content. When the secondary structure of β-LG is changed, epitope integrality could be destroyed, which might be helpful to understand the reduction in the allergenicity of β-LG.

### 3.3. Analysis of Intrinsic Fluorescence and UV Absorption Spectra

Fluorescence quenching is a process in the decrease in the intrinsic fluorescence of proteins by intermolecular interactions [27], and the change of protein fluorescence intensity can reflect the amino acid microenvironment of N-LG, U-LG, S-LG–LUT and U-LG–LUT. β-LG has inherent fluorescence properties due to the presence of aromatic amino acids, such as tryptophan (Trp), tyrosine (Tyr), and phenylalanine (Phe). As shown in Figure 1B, compared with N-LG, the maximum absorption peaks of S-LG–LUT and U-LG–LUT decreased and blue shifted, indicating that the amino acid microenvironment of β-LG changed. The Trp and Tyr residue of U-LG–LUT may be changed by ultrasound [28], indicating that ultrasound further promotes the formation of the LG–LUT complex.

UV-vis spectroscopy can be further used to determine whether β-LG interacts with LUT. At 278 nm, the tyrosine and tryptophan residues form the maximum adsorption peak. As shown in Figure 1C, compared with N-LG, the maximum absorption peak of U-LG significantly decreased, suggesting that ultrasound could destroy the stable conformation of β-LG. The maximum absorption peaks of S-LG–LUT and U-LG–LUT increased greatly and shifted blue [28], indicating that the interaction between β-LG and LUT resulted in changed in β-LG spatial structure. The chromophore of U-LG–LUT were further exposed, which was reflected by a higher absorption peak at 278 nm than that of S-LG–LUT. The results showed that ultrasound affected the formation of the LG–LUT complex.

### 3.4. Surface Hydrophobicity Analysis

Protein interface properties are related to surface hydrophobicity (H_0_). As shown in Figure 1D, the H_0_ values of U-LG, S-LG–LUT and U-LG–LUT were reduced, which is related to the decrease in non-polar amino acid exposure on the surface. The cavitation effect of ultrasonic wave can make part of β-LG accumulate, and the non-polar amino acids on the surface of some proteins are hidden inside the molecules [29], resulting in the decrease in the H_0_ value of U-LG. The decrease in the H_0_ values of S-LG–LUT and U-LG–LUT indicated that the polarity of the β-LG–LUT complex increased, which may be due to the interaction between LUT and hydrophobic amino acids on the β-LG surface, resulting in the blocked binding of ANS to the hydrophobic surface of β-LG [30]. In addition, the circular dichroism results show that the β-sheet content of U-LG, S-LG–LUT and U-LG–LUT increased, and the decrease in the H_0_ value is significantly correlated with the increase in protein β-sheet content, a result that is consistent with the secondary structure change of protein [31].

### 3.5. Molecular Docking

The results of molecular docking are shown in Figure 2. LUT binds to the β-sheet and β-turns regions near the α-helix on β-LG’s outer surface [11]. It has been reported that there are two main binding sites for specific hydrophobic ligands that bind β-LG. Site I is located in the central hydrophobic cavity of the β-barrel structure called the calyx. Site II is located on the outer surface of β-LG, and the main residues are Trp19, Tyr20, Tyr42, Gln44, Gln59, Gln68, Leu156, Glu157, Glu158 and His161 [11]. LUT bound with Thr18, Val43, and Leu156 through a hydrogen bond, which was in agreement with the results reported by Rahayu et al. [32]. Eight amino acid residues of β-LG (Trp19, Ser21, Tyr42, Glu45, Gln59, Glu158, Gln159 and His161) interacted with LUT through van der Waals bonds. Trp19, Leu156, Glu157, and His161 of β-LG, adjacent to the pocket, bind with LUT through strong hydrophobic interactions [25]. These results showed that hydrogen bonds, van der Walls bonds, and hydrophobic interactions were the major driving forces taking part in the formation of the β-LG–LUT complex.

### 3.6. IgG/IgE Binding Ability and Degranulation Analysis of KU812 Cells

The IgG binding ability of N-LG, U-LG, S-LG–LUT and U-LG–LUT were evaluated by indirect competitive ELISA. As shown in Figure 3A, the IC_50_ value of N-LG, U-LG, S-LG–LUT and U-LG–LUT were 11.208, 20.913, 42.68 and 119.317 μg/mL, respectively. The results show that β-LG bound via a non-covalent interaction with LUT could decrease allergenicity. The IC_50_ value of U-LG–LUT was the highest, implying that the ultrasound and non-covalent interaction treatment was a better desensitization method. The IgE binding ability of N-LG, U-LG, S-LG–LUT and U-LG–LUT were detected by indirect ELISA. The OD value represents the IgE binding ability of β-LG [22]. As shown in Figure 3B, the IgE binding ability of β-LG was N-LG > U-LG > S-LG–LUT > U-LG–LUT, results that were consistent with the changing trend of IgG binding ability.

The ability of cell degranulation to release IL-6 can be used as a key indicator of allergic reactions [33]. Histamine released by basophil-activated degranulation is a powerful inflammatory mediator, and the content of histamine is positively correlated with the severity of allergic reactions [34]. As shown in Figure 3C, among U-LG, S-LG–LUT and U-LG–LUT, the IL-6 release level was the lowest in U-LG–LUT. Histamine release followed a similar trend (Figure 3D). These results show that the β-LG–LUT complex during ultrasonic treatment inhibited the allergy reactivity of KU812 cells. In short, ELISA and cell experiment results show that β-LG binding with LUT via a non-covalent interaction could decrease the IgG/IgE binding activity and inhibit the allergy reactivity of KU812 cells.

### 3.7. Human Intestinal Microbiota Analysis

As shown in Figure 4, with the increase in sequencing quantity, the out of each group of samples increased and then tended to be flat, indicating that the sequencing data amount was sufficient to reflect the species information of the vast majority of microorganisms in the samples.

An alpha index analysis is performed on the sample, and the results are shown in Table 2. Compared with the control group, the Shannon index of N-LG decreased significantly, indicating that β-LG can reduce the diversity of human intestinal microbiota after simulating human gastrointestinal digestion in vitro. Compared with the N-LG group, the Shannon index values of S-LG–LUT and U-LG–LUT significantly increased, suggesting that LUT could alter the effect of β-LG on the decrease in human intestinal microbiota diversity. Compared with the Chao index in the control group, the N-LG group decreased significantly, and the S-LG–LUT and U-LG–LUT increased, indicating that the U-β-LG–LUT complex can increase the richness of human intestinal microbiota. According to the results of the Chao, Shannon and Simpson indices, the digested products of U-LG, S-LG–LUT and U-LG–LUT after simulating in vitro human gastrointestinal tract digestion can affect the community richness and diversity of human intestinal microorganisms.

Figure 5A shows that N-LG, U-LG, S-LG–LUT and U-LG–LUT have a different effect on the relative abundance of *Bacteroidota*, *Firmicutes*, and *Proteobacteria*. Compared with the dominant microbiota at the phylum level in the control group, N-LG significantly increased the relative abundance of *Bacteroidota* and decreased the relative abundance of *Firmicutes* (*p* < 0.05). This indicates that N-LG significantly changed the human intestinal microbiota. Compared with N-LG, the relative abundance of *Bacteroidota* of U-LG, S-LG–LUT and U-LG–LUT decreased, the relative abundance of *Firmicutes* and *proteobacteria* increased, and the change of U-LG–LUT was closer to that of the control group. The changes in richness and diversity of the dominant microbial microbiota may be related to food allergy [35].

As shown in Figure 5B, *Prevotellaceae*, *Ruminococcaceae*, *Lachnospiraceae*, *Selenomonadaceae* and *Bifidobacteriaceae* are the most predominant in the human fecal samples at the family level. The dominant genera in the control group are *Prevotellaceae*, *Ruminococcaceae*, Selenomonadaceae and *Bifidobacteriaceae*, while the dominant genera in N-LG, U-LG, S-LG–LUT and U-LG–LUT are *Prevotellaceae*, *Ruminococcaceae* and *Lachnospiraceae*, indicating that β-LG could affect the species of the dominant genera. Compared with the control group, the relative abundance of *Prevotellaceae* in N-LG and U-LG groups increased (*p* < 0.05), while the relative abundance of *Selenomonadaceae* and *Bifidobacteriaceae* decreased. Studies have shown that the abnormal composition of fecal microbiota is related to the occurrence of food allergy in infants [16]. The S-LG–LUT and U-LG–LUT groups alleviated the impact of the decline in the relative abundance of *Bifidobacteriaceae*, and the degree of mitigation was U-LG–LUT > S-LG–LUT. This may explain the fact that U-LG–LUT is the least allergic.

As shown in Figure 5C, *Prevotella, Faecalibacterium*, *Megamonas*, *Bifidobacterium* and *Collinsella* are the dominant bacteria at the genus level. Compared with the control group, the relative abundance of the *Megamonas* and *Bifidobacterium* of N-LG, U-LG, S-LG–LUT and U-LG–LUT decreased significantly, while the relative abundance of *Prevotella* increased significantly (*p* < 0.05), suggesting that β-LG could affect the composition of the human intestinal microbial community. Compared with the N-LG group, the relative abundance of *Bifidobacterium* in S-LG–LUT and U-LG–LUT increased (*p* < 0.05). These results indicate that S-LG–LUT and U-LG–LUT can alleviate the changes of human intestinal microbiota, and the degree of remission is U-LG–LUT > S-LG–LUT. These results show that ultrasound improved non-covalent modification and significantly reduced the antigenicity of β-LG.

*Bifidobacteria* are recognized probiotics, which can indirectly exert probiotic effects by the immunomodulatory regulation of intestinal cells or modifying the function of normal microbiota. In particular, the *Bifidobacterium* BB12 can effectively promote the balance of intestinal microbiota and improve human immunity [36]. Figure 6 shows the heat map analysis of the level of intestinal microbiota, and the red shades represent the relative abundance of the microbiota. Compared with the control group, the relative abundance of *Bifidobacterium* in the N-LG group decreased significantly. The S-LG–LUT and U-LG–LUT groups alleviated the decrease in the relative abundance of *Bifidobacterium* in the N-LG group, and the degree of relief was S-LG–LUT > U-LG–LUT. Combined with allergenicity analysis, the upregulation of *Bifidobacterium* can alleviate a β-LG-induced allergic reaction. Interestingly, U-LG–LUT showed a stronger desensitization ability than S-LG–LUT in allergenicity tests. The changes in relative abundance of *Bifidobacterium* may induce the change in β-LG allergenicity. Furthermore, the relative abundance of *Megamonas* also showed a significant difference in color depth between experimental groups.

### 3.8. Ultrasound Improved the Non-Covalent Interaction of β-Lactoglobulin with Luteolin: Effect on the Structure, Allergenicity and Human Intestinal Microbiota

β-LG is a major allergen in milk and dairy products, and non-covalent modification with small molecule compounds can reduce its allergenicity. In the current study, the β-LG–LUT complex under ultrasonic treatment not only reduced the IgG/IgE binding ability, and IL-6 and histamine release (Figure 3), but also has an effect on the human intestinal microbial ecosystem (Figure 5). The potential relationship between the reduction in allergenicity and the change of intestinal microbiota was analyzed.

The allergenicity of β-LG is closely related to the integrity of the allergic epitope. The conformational structures of the β-LG–LUT complex under ultrasonic treatment changed, which reflected by the decrease in α-helix content (Table 1), intrinsic fluorescence intensity (Figure 1B) and surface hydrophobicity (Figure 1D), whereas the β-sheet content (Table 1) and UV absorption intensity increased (Figure 1C). The main epitopes of β-LG include the peptides Trp19-Tyr20, Val43-Lys47, Leu57-Gln59, Cys66-Gln68 and β-corner (Leu149-ILE162) [11]. Combining the molecule dynamic simulation model with the conformational changes of proteins suggested that the binding process between β-LG and LUT induced by ultrasound was accompanied by changes in the epitopes [26]. It was shown in Figure 2A that LUT interacted with the residues Thr18, Tyr20, Val43, Glu44, Leu156, Glu157, and His161 of β-LG. Remarkably, some of these residues are located in the epitopes. Similar results were obtained in the molecular docking of oleic acid and β-LG [26]. When the original epitope was masked, the integrity of β-LG epitopes was destroyed, leading to a decrease in allergenicity. The result can explain the decrease in IgG/IgE binding activity (Figure 3A,B) and the mediator release (Figure 3C,D). Similar results were achieved in the study by Liu et al. [25], who proved that the changes in conformational epitopes could decrease allergenicity. Therefore, the destruction in the conformational epitopes of the β-LG–LUT complex dramatically reduced allergenicity.

The confusion of intestinal metabolic function caused by the imbalance of intestinal microbiota is closely related to the occurrence of allergic reactions [36]. Specific microbiota inhibit the cytokines produced by Th2 cells, reducing the release of inflammatory mediators and relieving food allergy symptoms [37]. After β-LG treatment in different ways, the intestinal microbiota composition had different changes. Compared with the N-LG group, the relative abundance of the *Bacteroidota* of the U-LG, S-LG–LUT and U-LG–LUT groups decreased and the relative abundance of *Firmicutes* and *Proteobacteria* increased (Figure 5A). The change trend of relative abundance of *Bacteroidota* showed a similar trend with ELISA and cell experiment results (Figure 3). It can be suggested that the changes in richness and diversity of the *Bacteroidota* may be related to food allergies. It was observed that the relative abundance of intestinal microbiota at the family level (Figure 5B) of β-LG modified by different treatments has a similar trend. Remarkably, we found the probiotic *Bifidobacteria* at the genus level (Figure 5C). *Bifidobacteria* are recognized probiotics, which can indirectly exert probiotic effects by the regulation of intestinal cells or modify the normal microbiota function. Although U-LG–LUT has the least allergic ability, the relative abundance of *Bifidobacteria* is not the largest, suggesting that *Bifidobacteria* is not the only intestinal microbe associated with allergenicity. Therefore, ultrasonic treatment coupled with LUT decreased the allergic ability of β-LG and altered human intestinal microbiota.

## 4. Conclusions

In this paper, the IgG/IgE binding capacity of sonicated β-LG–LUT complexes and their hypersensitivity in KU812 cells were significantly reduced, compared with LUT treatment alone, and also affected the composition of the human gut microbiota. The results of this study may be attributed to changes in the conformational epitope of β-LG. However, the evaluation of the relationship between β-LG allergenicity and intestinal microbiota composition is incomplete, and animal studies are needed to further confirm the mechanism.

## Figures and Tables

**Figure 1 foods-11-00988-f001:**
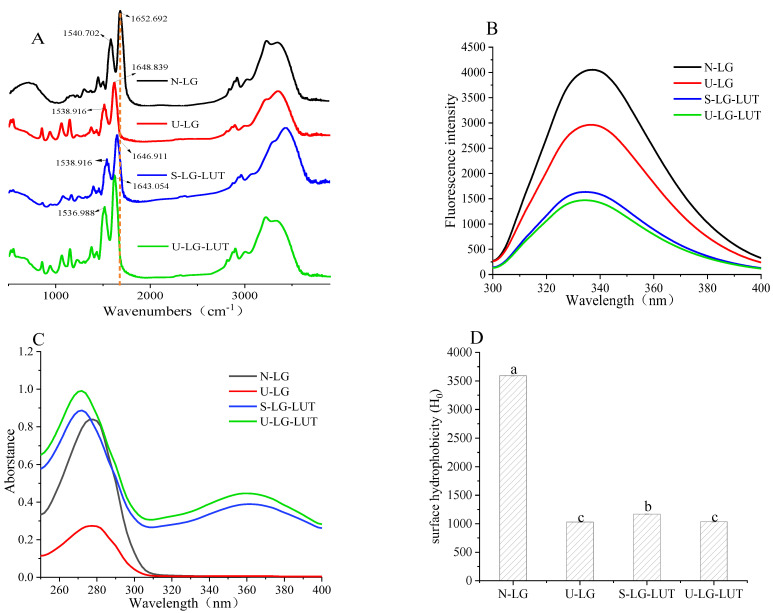
Infrared spectrum (**A**), intrinsic fluorescence intensity (**B**), UV absorption spectrum (**C**) and surface hydrophobicity (**D**) of N-LG, U-LG, S-LG–LUT and U-LG–LUT. Letters (a–c) in the bars mean significantly different (*p* < 0.05).

**Figure 2 foods-11-00988-f002:**
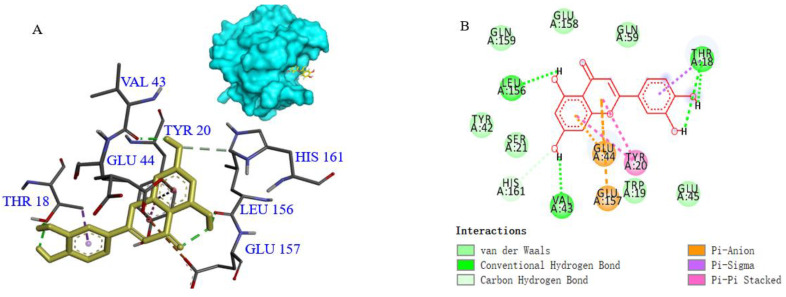
Binding interactions and 2D interaction diagrams of LUT binding with β-LG. The 3D interaction mode of β-LG and LUT (**A**), and the 2D plan view of the interaction between β-LG and LUT (**B**). In the 2D plan view, light green circles correspond to amino acid residues that interact with the LUT through van der Waals forces. Green, purple, and orange dashed lines represent hydrogen bonding, π-π accumulation, and π-cation interactions, respectively.

**Figure 3 foods-11-00988-f003:**
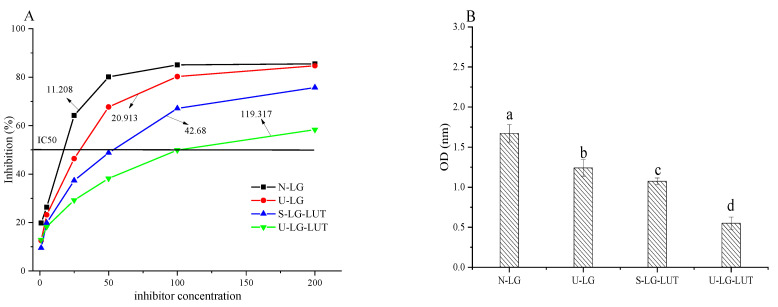
IgG (**A)** and IgE (**B**) binding ability, and the release of IL-6 (**C**) and histamine (**D**) of N-LG, U-LG, S-LG–LUT and U-LG–LUT. Letters (a–d) in the figure body mean significantly different (*p* < 0.05).

**Figure 4 foods-11-00988-f004:**
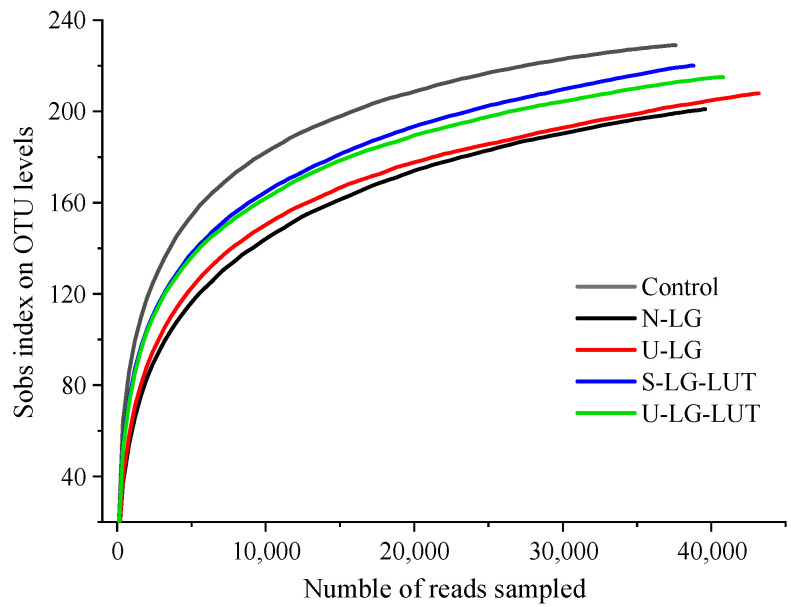
Dilution curves of high throughput sequencing results.

**Figure 5 foods-11-00988-f005:**
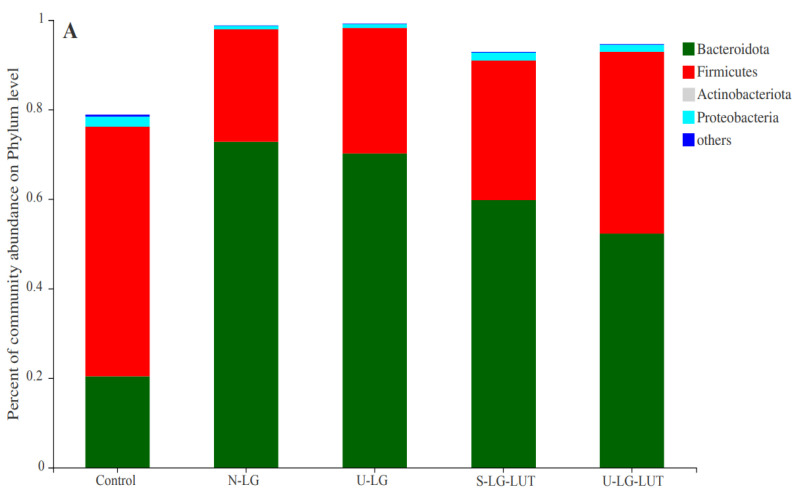
Effect of N-LG, U-LG, S-LG–LUT and U-LG–LUT on the community abundance at the phylum level (**A**), family level (**B**), and genus level (**C**); The one-way ANOVA bar plot of multi groups at the phylum level (**D**), family level (**E**), and genus level (**F**). * and ** symbols indicate significance *p* < 0.05 and *p* < 0.01.

**Figure 6 foods-11-00988-f006:**
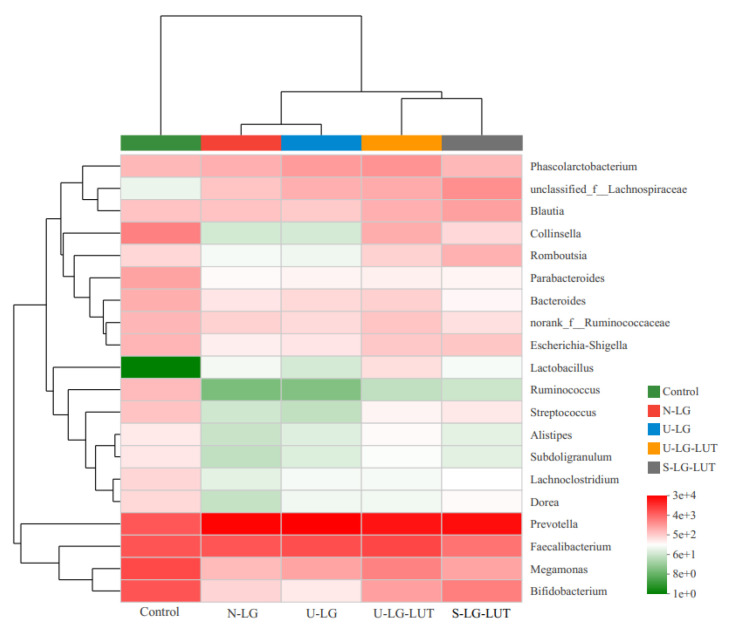
Effect of N-LG, U-LG, S-LG–LUT and U-LG–LUT on the community heatmap analysis at the genus level.

**Table 1 foods-11-00988-t001:** Secondary structure of N-LG, U-LG, S-LG–LUT and U-LG–LUT. Letters a–d in the table indicate significant differences (*p* < 0.05).

Sample	α-Helix (%)	β-Sheet (%)	β-Turn (%)	Random (%)
N-LG	13.24 ± 0.38 ^b^	29.14 ± 0.74 ^c^	22.96 ± 0.17 ^a^	34.64 ± 0.13 ^a^
U-LG	14.54 ± 0.20 ^a^	33.13 ± 0.31 ^b^	21.95 ± 0.23 ^a,b^	30.26 ± 0.36 ^c^
S-LG–LUT	2.39 ± 0.23 ^c^	42.37 ± 0.48 ^a^	22.33 ± 0.38 ^a,b^	32.90 ± 0.52 ^b^
U-LG–LUT	0.61 ± 0.12 ^d^	43.66 ± 0.76 ^a^	21.77 ± 0.71 ^b^	33.84 ± 0.81 ^a,b^

**Table 2 foods-11-00988-t002:** Intestinal microbiota index table (*p* < 0.05). Letters (a–c) in the table mean significantly different (*p* < 0.05).

Samples	Shannon	Simpson	Chao	Coverages
Control	3.21 ± 0.12 ^a^	0.09 ± 0.01 ^c^	252.03 ± 10.98 ^a^	0.99 ± 0.00 ^a^
N-LG	1.68 ± 0.17 ^c^	0.45 ± 0.06 ^a^	227.84 ± 13.17 ^b^	0.99 ± 0.00 ^a^
U-LG	1.96 ± 0.33 ^b,c^	0.38 ± 0.09 ^a,b^	252.26 ± 9.04 ^a^	0.99 ± 0.00 ^a^
S-LG–LUT	2.39 ± 0.02 ^b^	0.29 ± 0.01 ^b^	251.65 ± 3.20 ^a^	0.99 ± 0.00 ^a^
U-LG–LUT	2.32 ± 0.18 ^b^	0.29 ± 0.06 ^b^	237.41 ± 2.30 ^a,b^	0.99 ± 0.00 ^a^

## Data Availability

The data presented in this study are available upon request from the corresponding author.

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
