# Peer review of "Ultrasound Improved the Non-Covalent Interaction of β-Lactoglobulin with Luteolin: Regulating Human Intestinal Microbiota and Conformational Epitopes Reduced Allergy Risks"

_foods, 2022, doi:10.3390/foods11070988_

Round 1

Reviewer 1 Report

This an interesting study of the interaction of Luteolin and beta-LG. However, the  effect of ultrasound (US) seems not to be so clear described, and I am unsure about it effects. So this needs a more trorough  described

When applying US the temperature increase in the fluid, and this need to be measured in order to understand the denaturation of the protein.

And more literature comparation with heat denaturation of Beta-lg could also improve the paper

Specific comments:

Material and methods: In general info about the type of instrument use is lacking

L79 This is a Low concentration of Beta-LG. I doubt whether this is food or instant formula relevant.

L81-82. What kind of ultrasonic systems is used (E.g. brand, sonotrode system,  tip size). How much did the temperature increase, this is important according to whey protein denaturation

L159-171: The effect of US is not clear described

Figure 4 How much here is statistically significant and is one-way ANOVA the correct analysis here. Standard deviation should also be illustrated .

L259 However histamine is not lower than control- describe and explain

L340 is this the discussion section

Reviewer 2 Report

The concept of the research article is interesting. However, in the present form, it required substantial major revision. The major comments are as follows;

Introduction looks very general, in the introduction section, write the novelty of the work and the problem statement clearly. In order to the improvement of the quality content author should refer or cite recent articles related to the interaction of milk compounds. 

Section 3.1 (Infrared spectral analysis) should be rewritten as lot of grammar mistakes are there, also check in the whole manuscript.

Conclusion should be more detailed and brief.

Round 2

Reviewer 2 Report

Authors included all the suggested changes in the manuscript. Now it can be considered for publication